# ADAMTS-9 in Mouse Cartilage Has Aggrecanase Activity That Is Distinct from ADAMTS-4 and ADAMTS-5

**DOI:** 10.3390/ijms20030573

**Published:** 2019-01-29

**Authors:** Fraser M. Rogerson, Karena Last, Suzanne B. Golub, Stephanie J. Gauci, Heather Stanton, Katrina M. Bell, Amanda J. Fosang

**Affiliations:** 1University of Melbourne Department of Paediatrics, Royal Children’s Hospital, Parkville, Victoria 3052, Australia; fraser.rogerson@rmit.edu.au (F.M.R.); karena.last@mcri.edu.au (K.L.); sue.golub@mcri.edu.au (S.B.G.); heather.stanton@mcri.edu.au (H.S.); 2Murdoch Children’s Research Institute, Royal Children’s Hospital, Parkville, Victoria 3052, Australia; steph_pascoe@hotmail.com (S.J.G.); katrina.bell@mcri.edu.au (K.M.B.); 3Royal Melbourne Institute of Technology, 124 La Trobe Street, Melbourne, Victoria 3000, Australia

**Keywords:** aggrecan, aggrecanase, ADAMTS, cartilage, arthritis

## Abstract

A disintegrin and metalloproteinase with thrombospondin motifs (ADAMTS)-4 and ADAMTS-5 are the principal aggrecanases in mice and humans; however, mice lacking the catalytic domain of both enzymes (TS-4/5∆cat) have no skeletal phenotype, suggesting there is an alternative aggrecanase for modulating normal growth and development in these mice. We previously identified aggrecanase activity that (a) cleaved at E↓G rather than E↓A bonds in the aggrecan core protein, and (b) was upregulated by retinoic acid but not IL-1α. The present study aimed to identify the alternative aggrecanase. Femoral head cartilage explants from TS-4/5∆cat mice were stimulated with IL-1α or retinoic acid and total RNA was analysed by microarray. In addition to ADAMTS-5 and matrix metalloproteinase (MMP)-13, which are not candidates for the novel aggrecanase, the microarray analyses identified MMP-11, calpain-5 and ADAMTS-9 as candidate aggrecanases upregulated by retinoic acid. When calpain-5 and MMP-11 failed to meet subsequent criteria, ADAMTS-9 emerged as the most likely candidate for the novel aggrecanase. Immunohistochemistry revealed ADAMTS-9 expression throughout the mouse growth plate and strong expression, particularly in the proliferative zone of the TS-4/5-∆cat mice. In conclusion, ADAMTS-9 has a novel specificity for aggrecan, cleaving primarily at E↓G rather than E↓A bonds in mouse cartilage. ADAMTS-9 might have more important roles in normal skeletal development compared with ADAMTS-4 and ADAMTS-5, which have key roles in joint pathology.

## 1. Introduction

The weight-bearing properties of articular cartilage are conferred by the large and highly charged glycosaminoglycan side chains that are present on the proteoglycan aggrecan. In arthritic disease, the resilience of cartilage is compromised when large aggrecan aggregates are degraded by aggrecanases that are members of the ADAMTS (a disintegrin and metalloproteinase with thrombospondin motifs) family of zinc-dependent enzymes within the metzincin family of metalloproteinases. The principal aggrecanases thought to be responsible for aggrecan degradation are ADAMTS-4 [1] and ADAMTS-5 [2].

These enzymes were originally identified by their ability to cleave the aggrecan core protein at the TEGE ^373↓374^ ALGS bond in the aggrecan interglobular domain (IGD) [3,4,5] and at four additional sites within the chondroitin sulphate (CS)-rich region [6,7] (Figure 1) in response to stimulation with pro-catabolic agents, such as interleukin-1-alpha (IL-1α), IL-1β and retinoic acid. In contrast, ADAMTS-1 [8], -8 [9], -9 [10,11], -16 [11] and -18 [11] were thought to have only weak activity against aggrecan, in vitro and in vivo.

In the course of our studies, we identified a new aggrecanase activity that was present in cartilage explants from mice deficient in both ADAMTS-4 and ADAMTS-5 catalytic activity (TS-4/5∆cat mice) [13]. We found that retinoic acid treatment of TS-4/5∆cat cartilage explants did not stimulate aggrecanase activity at the classical TEGE ^373↓374^ ALGS or TAQE ^1572↓1573^ AGEG sites, but instead at the FREEE ^1467↓1468^ GLGS and SSELE ^1279↓1280^ GRGTI sites located in the chondroitin sulphate-rich region of the core protein (Figure 1). Thus, retinoic acid stimulates the expression of a novel aggrecanase activity in TS-4/5∆cat cartilage. This unexpected finding reveals that the substrate specificity of this unknown enzyme is distinct from ADAMTS-4 and -5. In addition, and in contrast to ADAMTS-4 and ADAMTS-5, the activity of the novel aggrecanase is stimulated by retinoic acid, but not by interleukin-1α (IL-1α) treatment [13].

In order to identify this novel aggrecanase, we cultured mouse femoral head cartilage explants from wildtype and TS-4/5∆cat mice. We used discrimination between the known effects of IL-1α and retinoic acid on the up-regulation of aggrecanase activity to screen for candidate extracellular proteinases whose mRNA expression levels were specifically increased by retinoic acid, but not by IL-1α treatment. Our results reveal that ADAMTS-9 is a major aggrecanase in TS-4/5∆cat mice, raising the possibility that ADAMTS-9 might be active in cartilage, in vivo, in the presence of inhibitors designed to block ADAMTS-4 and ADAMTS-5 activity.

## 2. Results

### 2.1. Retinoic Acid Regulates the Expression of a Novel Aggrecanase Activity via a Transcriptional Mechanism

We previously identified a novel aggrecanase activity that cleaves aggrecan in the CS-rich region in cartilage explants from mice deficient in both ADAMTS-4 and ADAMTS-5 [13]. Since this novel activity is stimulated by treatment with retinoic acid, but not IL-1α, we reasoned that the differential in susceptibility to retinoic acid and IL-1α could be used to identify aggrecanase candidates by microarray. We first confirmed that retinoic acid stimulates aggrecanolysis in TS-4/5∆cat cartilage via a transcriptional mechanism by culturing TS-4/5∆cat cartilage explants in the presence or absence of the transcriptional inhibitor, actinomycin D. Western blot analysis of the conditioned media showed that retinoic acid treatment stimulated the release of the aggrecan FREEE ^1467^ neoepitope from TS-4/5∆cat cartilage into conditioned medium (Figure 2a, lane 3), compared with the untreated control (Figure 2a, lane 1). However, the retinoic acid-induced increase in FREEE ^1467^ neoepitope was blocked in the presence of actinomycin D (Figure 2a, lane 4), confirming that the effects of retinoic acid in this system were indeed transcription-dependent. The chondrocytes remained viable in culture for a further 6 days in the presence or absence of retinoic acid, confirming that inhibition of aggrecan cleavage was not due to actinomycin-induced cell death.

### 2.2. Microarray and qPCR Studies Reveal ADAMTS-9 as a Strong Candidate for the Novel Aggrecanase

Based on our conclusion that retinoic acid exerts transcriptional control over aggrecanolysis in TS-4/5∆cat chondrocytes, we next used microarrays to shortlist genes encoding aggrecan-degrading proteinases, based on their up-regulation in response to retinoic acid but not IL-1α treatment. We focused on enzyme families known to cleave aggrecan, including the a disintegrin and metalloproteinase (ADAM), ADAMTS, matrix metalloproteinase (MMP), cathepsin and calpain families. The results presented as a volcano plot in Figure 2b show that only three genes matched these criteria: *Adamts-9*, *Mmp-11* and *Calpain-5* (Figure 2b). There was no increase greater than 1.5-fold in the expression of any ADAM or cathepsin gene in IL-1α-treated compared with retinoic acid-treated cultures.

To validate the microarray expression data, we used qPCR to measure the expression of key genes in cultured chondrocytes from TS-4/5∆cat mice. The results shown in Figure 3 confirm the up-regulation of mRNA expression for *Adamts-5* (Figure 3a), *Mmp-13* (Figure 3b) and *Mmp-11* (Figure 3d) by both IL-1α and retinoic acid, as reported previously, thus excluding these enzymes as the novel aggrecanase. The expression of *Calpain-5* was only weakly increased by IL-1α and retinoic acid (Figure 3c) and was therefore also considered not to be an aggrecanase. Overall, the mRNA data suggested that ADAMTS-9 was the strongest and most likely candidate for the novel aggrecanase. Figure 3e shows that while *Adamts-9* mRNA expression is increased by treatment with retinoic acid, there is no increase in *Adamts-9* gene expression induced by IL-1α. Similarly, retinoic acid, but not IL-1α, increased the expression of ADAMTS-9 protein detected as a band of approximately 200 kDa by Western blot (Figure 3f).

### 2.3. The Novel Aggrecanase is Inhibited by TIMP-3

Tissue inhibitor of metalloproteinase (TIMP)-3 is a potent endogenous inhibitor of ADAMTS and MMP family members [14]. We therefore used sensitivity to inhibition with TIMP-3 to further confirm the identity of the novel aggrecanase and found that when femoral head explants from TS-4/5∆cat mice were incubated with retinoic acid in the presence or absence of 500 nM TIMP-3, aggrecan cleavage at the FREEE ^1467↓1468^ GLGSV site was inhibited (Figure 4a, lane 5). An additional FREEE ^1467^ band migrating at approximately 50 kD and most likely the SSELE ^1279_1467^ FREEE fragment previously identified in human cartilage [15] was also generated in the presence of retinoic acid (Figure 4a, lanes 2 and 3) but blocked by high concentrations of TIMP-3 (Figure 4a, lane 5). As a control for the potency of TIMP-3 inhibition, cleavage at the MMP-sensitive DIPEN ^342↓343^ FFGVG site was also examined. TIMP-3 blocked cleavage at DIPEN ^342↓343^ FFGVG at the same high concentration (Figure 4b, lanes 4 and 5).

### 2.4. ADAMTS9 Expression and Aggrecanase Activity in the Growth Plate

Finally, we used immunohistochemistry to compare sites of ADAMTS-9 expression and aggrecanase activity in sections of the 3-week mouse growth plate. For both genotypes, FREEE ^1467^ neoepitope was present in selected regions of the growth plate including the proliferative zone (Figure 5a,b,e,f; P) and a thin layer of cells surrounding the secondary centre of ossification (Figure 5a,b,e,f; right-pointing arrow). The staining surrounding the secondary centre was more punctate in sections from TS-4/5∆cat mice (Figure 5e,f) than from wildtype mice (Figure 5a,b). FREEE ^1467^ neoepitope was also present at the metaphyseal border in wildtype mice (Figure 5a,b; left-pointing arrow) but was absent at this site in TS-4/5∆cat mice (Figure 5e,f). The different patterns of FREEE ^1467^ immunostaining at the metaphyseal border in wildtype mice, compared with TS4/5∆cat mice, suggest that FREEE ^1467^ immunoreactivity in the growth plate is a product of ADAMTS-4 and/or ADAMTS-5 activity. We therefore conclude that ADAMTS-9 is not essential for aggrecanase activity at the metaphyseal border.

Unlike the limited staining for the aggrecan FREEE ^1467^ neoepitope seen in wildtype growth plates (Figure 5 a,b,e,f), ADAMTS-9 immunostaining was strong throughout all zones of the wildtype growth plate, including the metaphysis (M), proliferative zone (P), hypertrophic zones (H) and in bony spicules (*diagonal dotted arrows*) protruding into the calcified region of the hypertrophic zone (Figure 5c,d). In contrast, there was no immunostaining for ADAMTS-9 in the hypertrophic/pre-hypertrophic zones (H) of the growth plate in TS4/5∆cat mice (Figure 5g,h). One interpretation of this unintuitive staining pattern is that ADAMTS-9 is not expressed in the hypertrophic zone (Figure 5g,h). However, a more likely interpretation is that the physical entrapment of residual, undegraded and highly charged aggrecan in the hypertrophic zone blocks diffusion of immunoreagents into this region of the sections, producing a false negative result.

## 3. Discussion

ADAMTS-9 is a multi-domain proteinase with key roles in the remodelling of extracellular matrices during growth and development [16,17,18,19,20,21]. Versican is a major substrate for ADAMTS-9, and in this study we show that the related proteoglycan, aggrecan, which is highly expressed in joint cartilage, is also a substrate for ADAMTS-9. Our results suggest that unlike ADAMTS-4 and ADAMTS-5, which are widely recognised as destructive aggrecanases in joint disease [22,23], the aggrecanolytic activity of ADAMTS-9 might be more important for normal joint development [18] than for pathology.

Previous studies in wildtype mice have reported the mRNA expression of ADAMTS-9 in chondrocytes and cartilage explants [20,21]; however, in the presence of ADAMTS-4 and ADAMTS-5 expression, it has not been possible to show direct evidence of ADAMTS-9 activity. Here, we have used TS-4/5∆cat mice to show that ADAMTS-9 is a bona fide aggrecanase with activity, not at the classical TEGE ^373↓374^ ALGS site in the IGD and TAQE ^1572↓1573^ AGEG site in the C-terminal domains, but instead ADAMTS-9 cleaves at FREEE ^1467↓1468^ GLGSV and SELE ^1279↓1280^ GRGT sites located centrally within the aggrecan core protein. These results extend previous studies, suggesting that ADAMTS-9 might also cleave at TAQE ^1771↓1772^ AGEG near the C-terminal G3 domain [10].

Our results are consistent with previous studies showing *Adamts-9* mRNA expression in neonatal growth plates [21] and in the proliferative and hypertrophic zones of the 7-week-old mouse growth plate [21]. In addition, we have used immunoreactivity against the FREEE ^1467^ neoepitope to show that TS-4/5∆cat mice express aggrecanase activity, and we propose that this activity is likely to be the product of ADAMTS-9 (Figure 5).

The distribution of FREEE ^1467^ neoepitope in the TS-4/5∆cat epiphysis suggests that ADAMTS-9 has a role in the remodelling of the secondary centre of ossification and the proliferative zone of the growth plate. However, whereas the absence of FREEE ^1467^ neoepitope at the metaphyseal border in TS4/5-∆cat mice suggests that ADAMTS-9 might not degrade aggrecan at this site, a more likely interpretation (and complication) of this staining pattern is that an accumulation of undegraded aggrecan in calcified regions of the hypertrophic zone impedes antibody penetration and compromises immunoreactivity.

Beyond the G2 domain, the aggrecan core protein is heavily decorated with glycosaminoglycan side chains. When ADAMTS-9 cleaves at FREEE ^1467↓1468^ GLGSV, the C-terminal third of aggrecan is lost from the cartilage, while the remaining N-terminal two thirds, immobilised in the cartilage by G1-domain binding to hyaluronan, provide the weight-bearing properties. Accordingly, cleavage at FREEE ^1467↓1468^ GLGSV is less deleterious for weight-bearing than cleavage in the IGD, which releases the entire CS-containing portion of aggrecan.

There are very few studies on the roles of regional aggrecanolysis. In vitro testing of cartilage explants showed that aggrecan loss mediated by IGD cleavage correlates with a loss of mechanical properties, whereas aggrecan loss due to cleavage at sites in the CS-2 domain does not appear to affect biomechanics [24]. In vivo studies to specifically explore the role of C-terminal domain cleavage have not been done, but it is widely held that C-terminal processing of aggrecan is part of normal homeostasis [25,26]. Furthermore, the proportion of C-terminally shortened aggrecan increases with age [27,28], suggesting that loss of the C-terminus is part of normal tissue maturation. Thus, ADAMTS-9 might have a role in maturation and normal cartilage homeostasis.

Alternatively, CS-2 domain cleavage might indeed have a role in cartilage pathology. In vitro evidence suggests that there is a hierarchy for aggrecanase cleavage and that cleavage in the IGD occurs after cleavage in the CS-2 domain [29,30,31]. If CS-2 domain cleavage is a prerequisite for IGD cleavage, then a CS-2 aggrecanase such as ADAMTS-9 might initiate an aggrecan-degrading cascade. Aggrecan fragments resulting from CS-2 domain cleavage are elevated in human synovial fluids following knee joint injury [32], raising the possibility that ADAMTS-9 contributes to the production of G3-containing fragments from injured cartilage. Furthermore, the G3 domain binds complement factors C1q and C3, thus activating the classical and alternative complement pathways [33], suggesting a possible role for ADAMTS-9 in complement regulation in inflamed joints.

## 4. Materials and Methods

### 4.1. Mice

The TS-4/5∆cat mice deficient in both ADAMTS-4 and ADAMTS-5 activity have been described previously [13]. The ∆-cat mice were bred from the TS-4 ∆-cat and TS-5 ∆-cat strains [13,22], which have in-frame deletions of exon 4 of the *Adamts-4* gene (TS-4 ∆-cat) and exon 3 of the *Adamts-5* gene (TS-5 ∆cat), encoding the respective catalytic domains [22]. Mutant ADAMTS-5 mRNA lacking exon 3 is translated and includes exons encoding the ADAMTS-5 C-terminal domain [34]. The TS-4/5∆cat mice are healthy with no abnormalities in skeletal growth or development. The Murdoch Children’s Research Institute Animal Ethics Committee approved all animal procedures under projects A605, A710, A795 and A868. Each experiment was done at least twice with a minimum of two biological replicates, except for the microarray which was done once with three mice per treatment.

### 4.2. Preparation of Primary Chondrocytes

Chondrocytes were isolated from the distal femoral and proximal tibial epiphyses of 6-day-old mice. The dissected epiphyses were digested for 1 h with 0.25% trypsin/EDTA to remove loose fibrous tissue, washed with PBS, then incubated overnight at 37 °C/5%CO_2_ with 300 U/mL collagenase II (Worthington Biochemical Corp, Lakewood, NJ, USA) in DMEM and 5% fetal bovine serum (FBS). The isolated cells were washed extensively with medium and seeded at 5 × 10^5^ cells/well in 24-well plates in DMEM, 10% FBS, 100 U/mL penicillin, 100 U/mL streptomycin, 2 mM l-glutamine and 20 mM HEPES. After 3 days in culture, the cells were washed, placed in serum-free DMEM and stimulated with either 10 ng/mL IL-1α (Peprotech, Rocky Hill, NJ, USA) or 10 µM retinoic acid (Sigma-Aldrich, St Louis, MO, USA) for a further 3 days. These concentrations of IL-1α and retinoic acid were previously confirmed by dose response to elicit maximal gene expression.

### 4.3. Culture of Femoral Head Cartilage Explants

Mouse femoral head cartilage explants were cultured as described previously [12,13]. Explants were incubated with 10 µM retinoic acid ± 10µg/mL actinomycin D (Life Technologies, Carlsbad, CA, USA) for 24 h. Conditioned medium was harvested, and the explants were washed and incubated with fresh retinoic acid for a further 6 days to confirm that actinomycin D had no adverse effect on chondrocyte activity or viability. To examine the inhibition of aggrecanase activity by tissue inhibitor of metalloproteinase-3 (TIMP-3), femoral head explants were co-cultured with 10 µM retinoic acid and 10–500 nM recombinant human TIMP-3 (a gift from Professor Hideaki Nagase, UK) and then analysed for aggrecanase products by Western blotting.

### 4.4. Western Blotting

To detect aggrecan neoepitopes, conditioned media from mouse femoral head cultures were dialysed against proteinase inhibitors and deglycosylated as described previously [34]. Deglycosylated samples were resolved by reducing SDS-PAGE on 4–20% gradient 7TGX gels (Bio-Rad, Hercules, CA, USA) and then transferred to polyvinylidene difluoride membranes. Aggrecan degradation was analysed using polyclonal antibodies [35] that recognise the FREEE ^1467^ neoepitope [36], created by aggrecanase cleavage in the CS-rich region, and the DIPEN ^341^ neoepitope [37], created by MMP cleavage in the IGD. To detect ADAMTS-9 protein, conditioned medium from primary chondrocyte cell cultures was blotted and detected with anti-ADAMTS-9 antibody from Abcam (Cambridge, UK), used at 1 μg/mL and detected with anti-rabbit-HRP (1:2000) from Dako (Jena, Germany). The volume of medium loaded was determined empirically for each neoepitope examined and results were normalised to the values for total aggrecan as described previously [12,13]. The anti-DIPEN and anti-FREEE antibodies were used at a concentration of 5 μg/mL.

### 4.5. Microarray Analysis

Femoral head cartilage explants harvested from 3-week-old TS-4/5∆cat mice were cultured for 3 days in serum-free DMEM containing 100 U/mL penicillin, 100 U/mL streptomycin, 2 mM l-glutamine and 20 mM HEPES. The explants were either (i) unstimulated, (ii) stimulated with 10 ng/mL recombinant human IL-1 or (iii) stimulated with 10 µM retinoic acid. These concentrations of IL-1α and retinoic acid stimulated maximal aggrecan loss from cartilage explants as described previously [13]. After treatment, the explants were snap frozen and stored at −80 °C. Three separate mice (unpooled) were included for each treatment. Total RNA from frozen, crushed cartilage samples was isolated using the RNeasy kit (Qiagen, Hilden, Germany) and the RNA quality was assessed by BioAnalyser. The microarray analyses were done under contract at the Australian Genome Research Facility using the Illumina Sentrix Mouse 1.1 chip. The data were analysed using the R limma package from Bioconductor, utilising the neqc function (quantile normalisation) with non-detected probes removed and a factor included for batch and biological replicates in the linear mode [38]. The microarray data is available at Gene Expression Omnibus (GSE110754).

### 4.6. Quantitative Real-Time Polymerase Chain Reaction (qPCR) Analysis of mRNA Expression

To validate the microarray data identifying the putative aggrecanases, mRNA expression of the candidates was analysed by real-time quantitative PCR (qPCR) with probes and primers from the Universal Probes Library (UPL) (Roche, Basel, Switzerland). The ∆∆Cp values derived from the Cp values determined by the Roche Light Cycler 480 were calculated for each gene and normalised against the geometric mean of the genes *Rpl10* and *Rpl26* [39], two reference genes specific for chondrocytes. The primer sequences used for mouse *Mmp13*, *Mmp11*, *Adamts5*, *Adamts9* and *calpain 5* genes are shown in Table 1.

### 4.7. Immunohistochemistry

Knee joints from 3-week-old mice were fixed in 4% (*w*/*v*) paraformaldehyde for 24 h and then decalcified in 7% EDTA/PBS, pH 7.4, for 3 weeks. Five-micron sections of paraffin-embedded epiphyses were treated with 3% (*v*/*v*) H_2_O_2_ to block endogenous peroxidase activity. Antigen retrieval included treatment with 10 mM citrate, pH 6.0, with 0.05% (*v*/*v*) Tween 20 for 30 min at 60 °C and then cooling to room temperature prior to incubation with 0.2% (*w*/*v*) hyaluronidase (Sigma) for 1 h at 37 °C. Non-specific antibody binding was reduced by treatment with 1.5% (*v*/*v*) normal goat serum. Sections were incubated with either rabbit anti-FREEE ^1467^ primary antibody (0.5 µg/mL), rabbit anti-ADAMTS-9 antibody (Abcam, Cambridge, UK; cat No 28279, 1:2000 dilution) or no primary antibody overnight at 4 °C. The following day, sections were incubated with HRP-labelled anti-rabbit IgG (Dako, Jena, Germany), visualised with diaminobenzidine and counterstained with haematoxylin.

### 4.8. Declarations Ethics Approval

The animal procedures used in this study were approved by the Murdoch Children’s Research Institute Animal Ethics Committee under ethics project numbers A605 (20 May 2008), A710 (18 April 2012) A795 (16 April 2015), A868 (17 May 2018).

### 4.9. Availability of Data and Materials

The data sets generated and analysed during this study are available in the GEO repository at accession number GSE110754.

## 5. Conclusions

The results of this study are significant in the context of continuing efforts to identify the physiological aggrecanases and to develop aggrecanase inhibitors for the clinical management of degenerative joint diseases. Overall, we show that in the absence of ADAMTS-4 and -5 activity, ADAMTS-9 is expressed and might therefore have a role in normal aggrecan remodelling during skeletal growth and development. While our findings suggest that ADAMTS-9 has important roles in skeletal growth, it is not clear what role(s), if any, ADAMTS-9 might have in pathological aggrecanolysis. In future work, it will be important to determine whether aggrecanase inhibitors targeting ADAMTS-4 and/or ADAMTS-5 might also affect ADAMTS-9 activity in adult tissues or influence endochondral ossification in developing and juvenile skeletal tissues.

## Figures and Tables

**Figure 1 ijms-20-00573-f001:**
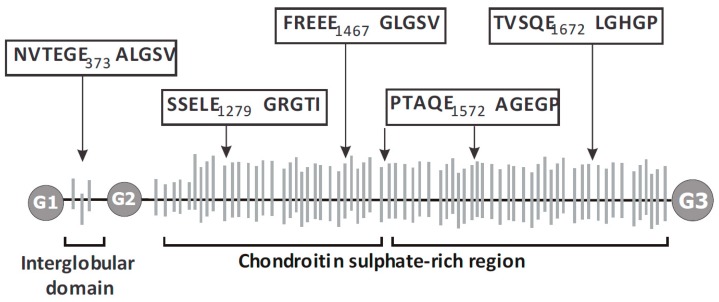
Aggrecanase cleavage sites and neoepitopes in the mouse aggrecan core protein. The domain structure of the mouse aggrecan core protein, with globular G1, G2 and G3 domains and intervening interglobular and chondroitin sulphate-rich regions are shown. The amino acid sequences flanking the conserved cleavage sites are shown in boxes with amino acid numbers at the P1 position. Modified from [12].

**Figure 2 ijms-20-00573-f002:**
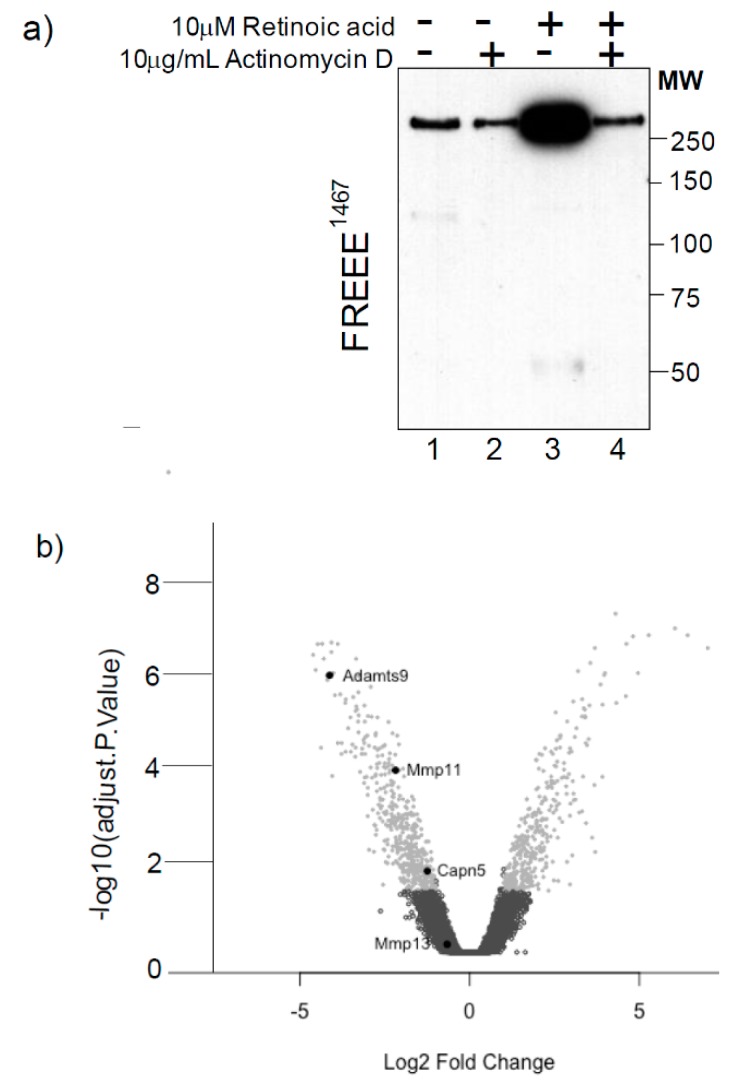
Identifying candidate aggrecanases in TS4/5∆cat mice. (**a**) Aggrecanase cleavage in TS-4/5∆cat femoral head cartilage explants, cultured untreated (lane 1) or in the presence of actinomycin D (lane 2), retinoic acid (lane 3) or both actinomycin D and retinoic acid together (lane 4) was analysed by FREEE ^1467^ neo-epitope Western blotting. (**b**) The results of the microarray studies, represented as a volcano plot, identify candidate proteinases for the novel aggrecanase.

**Figure 3 ijms-20-00573-f003:**
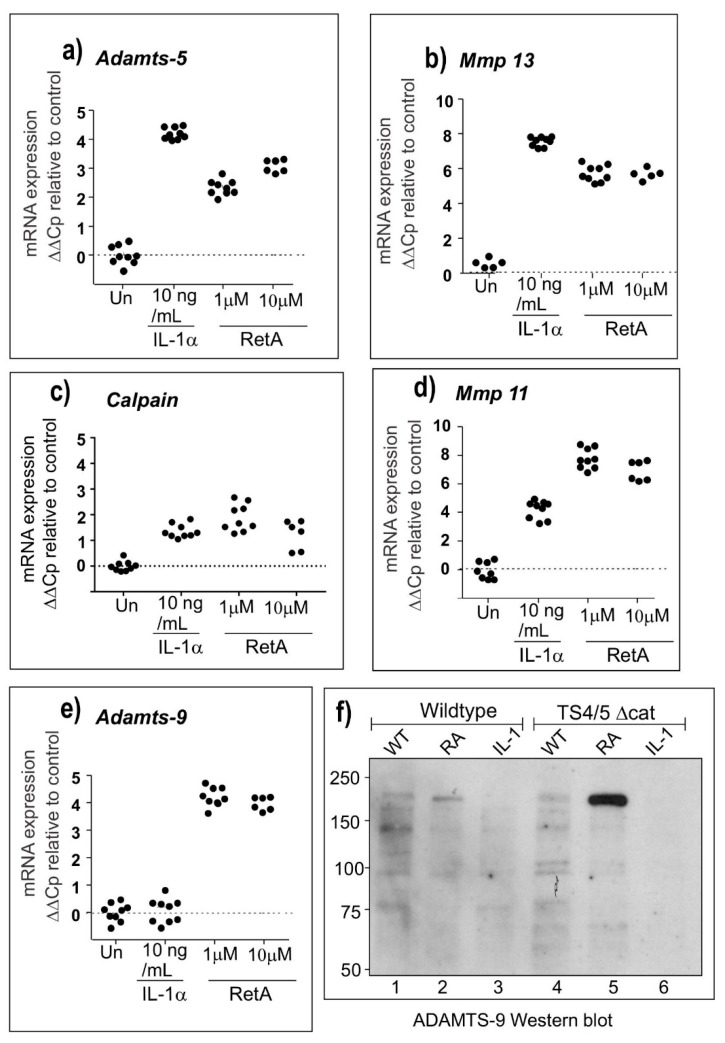
q-PCR and Western blot analyses of candidate aggrecanases. Candidate aggrecanases expressed in cultured cartilage explants were analysed by qPCR for genes including *Adamts-5* (**a**), *Mmp-13* (**b**), *Calpain* (**c**), *Mmp-11* (**d**), and *Adamts-9* (**e**). Protein extracts from Wildtype and TS4/TS5Δcat cartilage extracts were also analysed for ADAMTS-9 expression by Western blotting (**f**).

**Figure 4 ijms-20-00573-f004:**
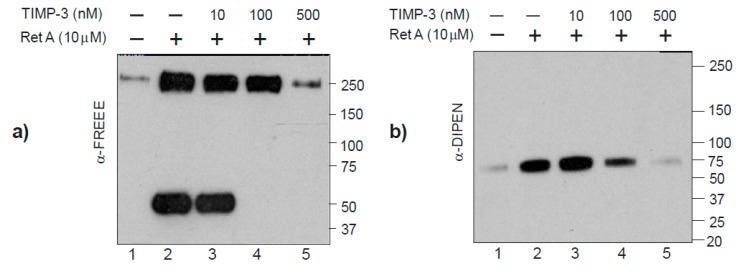
The novel aggrecanase is inhibited by tissue inhibitor of metalloproteinase (TIMP)-3. Western blot analyses show that TIMP-3 inhibits retinoic acid-induced aggrecanolysis at FREEE ^1467↓1468^ GLGSV (**a**) and DIPEN ^341↓342^ FFGVG (**b**) in TS-4/5Δcat cartilage explants.

**Figure 5 ijms-20-00573-f005:**
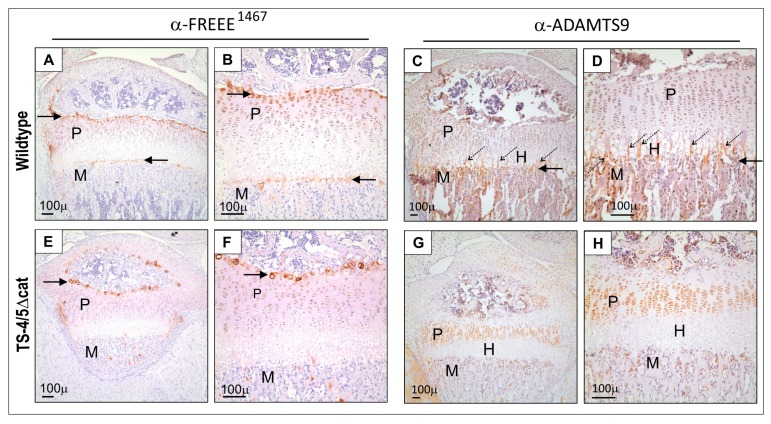
ADAMTS-9 protein and FREEE ^1467^ neoepitope are present in the 3-week-old mouse joint. M, metaphysis; P, proliferative zone of the growth plate; left-pointing arrow, metaphyseal border; right-pointing arrow, boundary of the secondary centre of ossification; diagonal dotted arrows, columns of calcified cartilage; H, hypertrophic zone of the growth plate. Panels (**a**,**b**,**e**,**f**) show staining for the FREEE ^1467^ neoepitope. Panels (**c**,**d**,**g**,**h**) show staining for ADAMTS-9 protein. Panels (**a**–**d**) show cartilage from wildtype mice. Panels (**e**–**h**) show cartilage from TS-4/5Δcat mice.

**Table 1 ijms-20-00573-t001:** Primer sequences for qPCR analyses.

Gene	Sequence
*Mmp-13* F	TCAAGGCTATGCACACTGGT
*Mmp-13* R	CACTATGGTCTTTTCAATGCCTAA
*Mmp-11* F	CAGGCCAAAAGGTACACAGC
*Mmp-11* R	CCTTTGAGGTTCCGTGTCTC
*Adamts-5* F	ATGCAGCCATCCTGTTCAC
*Adamts-5* R	CATTCCCAGGGTGTCACAT
*Adamts-9* F	ACAGCCATCAGAGAGTGCAA
*Adamts-9* R	TCCTACACAGTACTTCCCACCAT
*Calpain 5* F	CTAGCCTCCGCTCCAGTG
*Calpain 5* R	AAGAAGGGGAGGCACCTG

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
