# Peer review of "ADAMTS-9 in Mouse Cartilage Has Aggrecanase Activity That Is Distinct from ADAMTS-4 and ADAMTS-5"

_ijms, 2019, doi:10.3390/ijms20030573_

Reviewer 1 Report

The author investigated the function of ADAMTS-9 in mouse cartilage and compared with ADAMTS-4 and -5. They found ADAMTS-9 has a novel specificity for aggrecan, cleaving primarily at EG, rather than EA bonds in mouse cartilage. 

This study was designed based on their previous findings that aggrecanase activity in femoral head cartilage explants from mice deficient in both ADAMTS-4 and ADAMTS-5 catalytic activity. They found retinoic acid-treatment of TS-4/5deltacat cartilage explants stimulated aggrecanase activity. Because their previous reports were established, this manuscript is next story from their previous one. Their conclusion is ADAMTS-9 might have more important roles in normal skeletal development, compared with ADAMTS-4 and ADAMTS-5 which have key roles in joint pathology.

Data is clear and I can agree that this manuscript is acceptable. But, although this manuscript's data is clear, overall story flow-through in introduction must be improved. So, this reviewer is asking to author to describe more detailed information of general ADAMTSs in introduction part. 

Author Response

1.      I have lengthened the Introduction to improve the story flow.

2.      The duplicated legends for figures 2 and 3 have been deleted.

3.      I have reformatted the references. During the revision of the manuscript, reference 12 was removed, and the TIMP-3 reference added. I noted that my Endnote software had not renumbered the references when I moved the Methods section to the end of the manuscript, as per IJMS guidelines. Therefore I have hand numbered all the references using track changes. All citations are now in ‘[]’ brackets.

4.      I have added the project numbers for the Animal Ethics projects in lines 254, 341 and 342.

Reviewer 2 Report

Rogerson and colleague conducted microarray analysis and identified ADAMTS-9 which has aggrecanase activity to cleave at E-G bounds in aggrecan core protein. The expression of ADAMTS-9 is upregurated by retinoic acid in vitro and histological analysis revealed ADAMTS-9 expression throughout mouse growth plate. Histological analysis also suggests distinct function of ADAMTS-9 from that of ADAMTS-4 and ADAMTS-5 which are known crucial aggarecanases in joint pathology. Overall, the study was well designed, the results are convincing and the authors statement is reasonable. I have a few minor comments.

1.     In line 119, the authors wrote following sentence without any references: “TIMP-3 is a potent endogenous inhibitor of ADAMTS and MMP family members.” The authors need to add references for this.

2.     Figure legends for figures 2 and 3 are duplicating. The authors need to revise it.

3.     In figures 3f, 4a and 4b, labeling of y-axis, probably molecular weight, is missing. The authors need to revise this.

4.     What are sources of IL-1 alpha and retinoic acid in growth plate? Why did the authors focus on these two factors? Additional explanation will make this article reader-friendly.

Author Response

I have revised ijms-420316 (attached), with changes tracked in MSWord. Details of the changes are below.

1.      I have lengthened the Introduction to improve the story flow.

2.      The reference for TIMP-3, requested by Reviewer 2, is in line 149 (reference #14).

3.      The duplicated legends for figures 2 and 3 have been deleted.

4.      I have reformatted the references. During the revision of the manuscript, reference 12 was removed, and the TIMP-3 reference added. I noted that my Endnote software had not renumbered the references when I moved the Methods section to the end of the manuscript, as per IJMS guidelines. Therefore I have hand numbered all the references using track changes. All citations are now in ‘[]’ brackets.

5.      I have added the project numbers for the Animal Ethics projects in lines 254, 341 and 342.

I have not made two of the changes requested by Reviewer 2.

·        I have not added ‘MW’ to the y-axis in Figures 3f, 4a and 4b. This is difficult to do using track changes and I think it is unnecessary to do so. However, I am happy to provide new figures with ‘MW’ included, as separate files, if you require it.

·        I have not listed the sources of endogenous IL-1alpha and retinoic acid in the growth plate in the manuscript. We used exogenous IL-1alpha and retinoic acid as tools to stimulate cells and cartilage in culture. By discriminating between which enzymes were regulated by IL-1alpha and which were regulated with retinoic acid, we were able to track the novel aggrecanase. I have made this point more clearly in the revised introduction (see new text at lines 85-89).